# A Narrative Review of the Roles of Nursing in Addressing Sexual Dysfunction in Oncology Patients

**DOI:** 10.3390/curroncol32080457

**Published:** 2025-08-14

**Authors:** Omar Alqaisi, Suhair Al-Ghabeesh, Patricia Tai, Kelvin Wong, Kurian Joseph, Edward Yu

**Affiliations:** 1Faculty of Nursing, Al-Zaytoonah University of Jordon, Airport Street, 11118 Amman, Jordan; suhair_alghabeesh@yahoo.com; 2Department of Radiation Oncology, University of Saskatchewan, Saskatoon, SK S7N 5A2, Canada; ptai2@yahoo.com; 3Astellas Pharma Canada, 675 Cochrane Drive, Suite 650 West Tower, Markham, ON L3R 0B8, Canada; kelvin_wong@astellas.com; 4Department of Radiation Oncology, University of Alberta, Cross Cancer Center, Edmonton, AB T6G 2G5, Canada; kurian.joseph@albertahealthservices.ca; 5Department of Radiation Oncology, Western University, London, ON N6A 3K7, Canada; eyu@uwo.ca

**Keywords:** cancer, sexual care, nursing, immunotherapy, targeted therapy, sexual dysfunction, menopause, body image, erectile dysfunction, ejaculatory disorders

## Abstract

Sexual challenges are common yet often unaddressed issues for patients undergoing cancer treatment, significantly impacting their overall well-being and relationships. This review aimed to demonstrate how nurses can help with these sensitive concerns, what difficulties they face, and what solutions can improve care. We found that while nurses are crucial in providing support, they often lack specific training, face cultural discomfort, and are limited by hospital policies. Our findings suggest that better education for nurses, clear guidelines, and teamwork among healthcare providers are essential. By making sexual health a regular part of cancer care, we can significantly enhance the quality of life for cancer survivors and ensure a more complete approach to their recovery. This work provides valuable guidance for various healthcare professionals to support cancer patients.

## 1. Introduction

Sexuality is recognized as an essential and natural part of everyone’s life, regardless of age or physical condition [1], and is considered a central aspect of healthcare [2]. The harmful impacts of cancer and its treatments on sexuality are well known [3,4,5,6]. Patients with cancer may experience a range of sexual health challenges, including decreased libido, negative body image, vaginal dryness, genital hypoesthesia, negative body image, pain during sexual intercourse, and difficulty achieving orgasm [7,8].

The prevalence of this problem is very high. For example, a recent systematic review and meta-analysis of women with cancer revealed that up to 66% experienced sexual problems, with rates ranging from 58% for colorectal cancer to 55–89% for breast and gynecological cancers [9]. Additionally, 77.5–88.7% of breast cancer patients in China experience sexual health problems [10]. Furthermore, Proctor et al. reported that 57.5% of cancer survivors reported scores indicative of sexual dysfunction, with a similar prevalence observed among both men and women [11]. The WHO recommends patient-centered sexual healthcare, which delivers proper education together with counseling and medical support to individuals [12]. There is a lack of standardized protocols for incorporating sexual health education into routine cancer care practices in many institutions [13].

Sexual problems significantly affect patients’ physical and emotional well-being and quality of life and even lead to marital breakdown and suicide during the long cancer journey [14,15,16]. Sexual dysfunction is a common issue among cancer patients and has a significant effect on their overall well-being. However, research shows that it is frequently overlooked and rarely addressed in clinical settings [17,18]. Many patients do not actively seek medical support for sexual health concerns. For example, a study in mainland China revealed that although 42% of women with breast cancer recognized the value of receiving sexual health information from oncology professionals, 76% had never consulted healthcare providers about these issues [18]. Instead, they often turn to friends, peers, or online sources, which may lack reliability or effectiveness [19]. In clinical practice, healthcare professionals such as oncologists and nurses tend to prioritize treatment, rehabilitation, nutrition, and symptom management while often neglecting sexual health [20].

Research into nurses’ perspectives on addressing patients’ sexual health indicates that systemic challenges within the biomedical model of care hinder adequate attention to this issue. These challenges include short appointment times, insufficient training, and unclear professional responsibilities [21]. Additionally, nurses often frame sexual health in purely functional terms, leading to insufficient screening, communication, and interventions related to sexual issues [22]. Therefore, this narrative review seeks to examine existing research on the role of nurses in recognizing, addressing, and supporting sexual dysfunction in cancer patients.

## 2. Materials and Methods

This narrative review aimed to explore the roles of nurses in addressing sexual dysfunction among oncology patients. A comprehensive literature search was conducted across five major academic databases, namely, PubMed, CINAHL (Cumulative Index to Nursing and Allied Health Literature), Scopus, Web of Science, and ScienceDirect, spanning the period from January 2010 to May 2025.

To ensure inclusivity and depth in the search strategy, Boolean operators (AND, OR) were used to combine a broad set of keywords. These included sexual health, cancer, nursing, nursing roles, immunotherapy, targeted therapy, sexual dysfunction, vaginal dryness, genitourinary syndrome of menopause, sexual desire, body image, erectile dysfunction, climacturia, ejaculatory disorders, dyspareunia, and oncology.

A wide range of publication types was considered to capture diverse perspectives and insights. These included original research articles, review papers, case reports, randomized and nonrandomized controlled trials, cohort studies, cross-sectional studies, and surveys. However, the primary focus remained on original research studies that specifically addressed the role of nurses in managing sexual dysfunction in cancer patients.

Explicit inclusion and exclusion criteria were applied to refine the scope of the review. Articles were included if they were published in English and if they directly discussed nursing interventions or roles related to sexual dysfunction in oncology populations. The exclusion criteria were as follows:Non-English publications;Conference abstracts and commentaries;Studies not directly addressing the core topic;Duplicate articles.

The initial search yielded 1245 records. After 235 duplicates were removed, 1010 titles and abstracts were screened. On the basis of the eligibility criteria, 50 articles were ultimately selected for inclusion in the final review. Discrepancies in article selection between the two lead reviewers (OA and SA) were resolved through consultation with a third reviewer (PT), maintaining the rigor and objectivity of the review process.

For all the selected studies, the following relevant data were systematically extracted: study aims, sample size and characteristics, geographical location, data collection and analysis methods, and key findings. To ensure the inclusion of the most recent literature, alerts were set up in the databases to capture newly published relevant studies during the review period.

A thematic synthesis and the IMRAD (Introduction, Methods, Results, and Discussion) approach [23] were employed to categorize and interpret the data, enabling the identification of recurring themes, current challenges, and emerging trends in the nursing management of sexual dysfunction among cancer patients. Additionally, backward citation tracking (snowballing) was employed by reviewing the reference lists of relevant articles, helping to uncover additional studies that may have been missed in the original search.

The insights gained through this comprehensive review were critically evaluated and integrated into the main body of the manuscript, providing both a broad overview and a detailed interpretation of the current evidence landscape on this important topic.

## 3. Results

A total of 50 studies met the inclusion criteria and were reviewed. The findings are organized into five thematic areas: (1) prevalence and impact of sexual dysfunction in cancer patients, (2) psychological and social impact of sexual dysfunction, (3) barriers to addressing sexual dysfunction in cancer care, (4) interventions for managing sexual dysfunction in cancer patients, and (5) nurses’ attitudes and beliefs about sexual health. Table 1 and Figure 1 provide a consolidated summary of these themes, highlighting key nursing-related interventions and barriers identified in each area.

### 3.1. Prevalence and Impact of Sexual Dysfunction in Cancer Patients

A detailed summary of the prevalence of sexual dysfunction, affected populations, and associated symptoms across different cancer types is provided in Table 2. In 2022, there were an estimated 20 million new cancer cases and 9.7 million cancer-related deaths globally [46]. While survival rates for many cancers have improved, the long-term consequences of cancer treatments, including sexual dysfunction, remain poorly understood [46]. Studies suggest that cancer survivors often experience long-term difficulties with intimacy, body image, and relationship satisfaction due to sexual dysfunction [46]. Despite the recognition of sexuality as a core component of quality of life, sexual health interventions remain inconsistent across healthcare systems worldwide [47]. The WHO advocated integrating sexual health assessments into cancer survivorship programs to ensure that patients receive comprehensive support that addresses both physical and psychological well-being [48].

The prevalence of sexual dysfunction among cancer patients depends on their specific cancer diagnosis; the treatments they have received; and personal health factors such as age, sex, and pre-existing medical conditions [49]. In addition to direct physical symptoms, sexual dysfunction has also had a profound psychological effect on survivors. Physical changes and psychosocial challenges often lead to anxiety, depression, and self-consciousness, causing a loss of confidence and avoidance of intimacy [50,51]. However, survivors typically receive minimal information or support for managing these sexual side effects, resulting in prolonged distress and a diminished quality of life [28,29]. This lack of guidance exacerbates their difficulties, further reducing relationship satisfaction and negatively impacting mental health outcomes [31]. Despite how common and impactful these issues are, many patients remain reluctant to seek professional help for sexual dysfunction [24,25]. Feelings of embarrassment, stigma, or the assumption that sexual health is not a priority in cancer often discourage open discussion of these problems. Moreover, healthcare providers frequently lack the training, confidence, and institutional support to address sexual issues proactively in oncology practice [52,53]. Together, these factors create a significant gap in care delivery in regard to managing the sexual health of cancer survivors. Closing this gap requires a coordinated, multidisciplinary approach.

Experts advocate implementing routine sexual health assessment as part of cancer follow-up and providing healthcare professionals with dedicated training and resources to manage cancer-related sexual problems, thereby making sexual well-being an integral component of standard survivorship care [52,53]. By ensuring that sexual health is fully integrated into survivorship programs, healthcare systems can better support the long-term quality of life of cancer survivors (Table 2).

**Table 2 curroncol-32-00457-t002:** Summary of the prevalence of sexual dysfunction among cancer patients, broken down by cancer type and gender.

References	Cancer	Gender	Reported Prevalence	Common Manifestations/Concerns
[26,54,55,56]	Breast	F	16–100% (majority 50–75%)	Sexual difficulty, sexual pain, altered sexual self-image, reduced desire, vaginal dryness, dyspareunia
[27,28,56,57]	Gynecologic	F	30–100%	Vaginal dryness, dyspareunia, loss of sensation, decreased libido, hot flashes, difficulty achieving orgasm, and painful intercourse
[56,58]	Rectal (surgery)	F, M	19–62% (most ~60%)	Sexual dysfunction, body image concerns (stomas), and relationship issues.
[41,42,56]	Prostate	M	40–100%	Erectile dysfunction, loss of desire, problems reaching orgasm
[59]	Childhood cancer	F, M	~1/3 overall, ~½ with 1+ problem, 30% with 2+ problems	Difficulties relaxing/enjoying sex/becoming sexually aroused/achieving orgasm, erection difficulties (men), altered body image, fertility concerns
[60,61]	Hematologic (e.g., lymphoma)	F, M	54% decreased activity, 41% decreased interest	Decreased sexual activity, decreased sexual interest
[56]	Lung	F, M	~50% loss of libido (overall), ~40% decrease in sexual activity in F)	Loss of libido, decrease in sexual activity
[56]	Head and neck	F, M	24–100% negative impact	Negative impact on sexuality

F: female; M: male.

### 3.2. Psychological and Social Impact of Sexual Dysfunction

#### 3.2.1. Impact on Partners

Cancer-related sexual dysfunction affects not only patients but also their partners. Multiple studies have reported that partners often experience significant emotional distress, loneliness, and anxiety as they attempt to support their loved one through cancer and its sexual health challenges [29,62]. These partners frequently face a dual burden: concerns for the patient’s well-being and the strain of unmet intimacy needs. Qualitative evidence indicates that male partners of women with cancer, for example, describe feelings of isolation and a profoundly altered sexual relationship, underscoring the partner’s vulnerability in this context [29]. These findings highlight the need for partner-inclusive support, as addressing a couple’s needs (not just patients) is crucial to mitigating the psychosocial toll on partners and maintaining healthy relationship dynamics [29,62].

#### 3.2.2. Psychological Consequences

Untreated individuals with sexual dysfunction can severely undermine the mental health of cancer survivors. Studies have linked sexual problems with elevated levels of psychological distress, notably increased anxiety, depressive symptoms, and reduced self-esteem, among cancer patients [32,33]. As sexual function declines or side effects (e.g., body image changes, pain) emerge, patients often report feeling a diminished sense of self and femininity/masculinity, which can lead to social withdrawal and isolation [32,33]. These psychological consequences are not isolated findings; instead, they appear consistently across diverse cancer populations. The emotional burden of sexual dysfunction is profound and pervasive, suggesting that survivors require not only medical treatment but also proactive psychosocial support to address issues such as fear, frustration, and loss of identity that accompany sexual health changes.

#### 3.2.3. Relational Strains

Sexual dysfunction frequently places strain on intimate relationships. Research indicates that a decline in sexual intimacy and satisfaction can spark frustration, miscommunication, and conflict between partners, thereby eroding overall relationship quality [24,30]. In particular, couples who avoid discussing sexual issues tend to experience greater marital dissatisfaction. Communication about sexual health is often lacking, which exacerbates misunderstanding and emotional distancing [24,30]. Over time, this pattern of poor communication and unmet needs can lead to serious relational consequences. Some studies have documented cases of deteriorating partners’ relationships or even separation linked to unaddressed sexual problems in survivorship [20,30]. Conversely, couples who manage to maintain an honest dialog and seek help for sexual concerns typically fare better, highlighting communication as a key moderating factor. The literature, therefore, emphasizes that fostering healthy communication and providing counseling for couples is essential to prevent sexual dysfunction from translating into a long-term relational breakdown.

#### 3.2.4. Coping Mechanisms

Despite these challenges, sexuality remains fundamental to many patients’ quality of life and identity [31]. Survivors and their partners employ various coping mechanisms to navigate the sexual changes imposed by cancer and its treatment. Some individuals adopt personal coping strategies, such as mindfulness and other stress reduction techniques, which have been associated with lower distress and better emotional adjustment in the face of cancer-related sexual problems [61]. Engaging in psychosocial support, such as joining support groups or couples’ therapy, is another coping avenue that can help normalize conversations about sexual health and reduce feelings of isolation. In addition, experts underscore that effective coping often requires external intervention: comprehensive strategies that address both the physical and psychological aspects of sexuality are advocated as best practices [63,64]. Integrative intervention (combining medical management of symptoms with counseling or sex therapy) has been recommended to bolster patients’ and partners’ coping capacity, enabling them to adapt to sexual changes while preserving emotional intimacy [63,64].

Across the reviewed studies, sexual dysfunction in cancer patients has emerged as a multidimensional problem with consistently observed psychological and social ramifications. Numerous publications spanning different cancer types and cultural contexts document the same core issues: heightened emotional distress and interpersonal difficulties resulting from untreated sexual health concerns [24,32,33]. Notably, earlier works draw attention to the prevalence and seriousness of these psychological impacts (for instance, raising awareness that sexuality should not be overlooked even in life–threatening illnesses) [33]. More recent research has shifted toward identifying protective factors and solutions, such as mindfulness-based coping and holistic intervention programs, reflecting a growing focus on mitigation rather than merely describing the problem [61,63]. This trend signifies an important evolution in the literature: from recognizing that sexual dysfunction profoundly affects patients and their partners to actively exploring ways to buffer its impact. Collectively, the evidence underscores that integrating psychosocial support into sexual health is essential for improving cancer survivors’ quality of life and maintaining healthy relationships in survivorship [63,64].

### 3.3. Barriers to Addressing Sexual Dysfunction in Cancer Care

Despite growing awareness of the importance of sexual well-being in cancer survivorship, multiple barriers continue to hinder its integration into oncology nursing care. These challenges arise at various levels, including personal discomfort and limited communication skills among nurses [29,34,35], cultural and religious taboos that discourage open dialog [36,65], institutional constraints such as the absence of protocols and time pressure [24,30,38,39,40,66], and patient-related factors such as embarrassment or uncertainty about discussing sexual concerns [24,62,65,67]. As summarized in Table 3, these barriers interact to create systemic silence around sexual dysfunction, limiting nurses’ ability to address this critical aspect of holistic cancer care.

### 3.4. Interventions for Managing Sexual Dysfunction in Cancer Patients—Training Programs to Build the Trust of Patients and Achieve Holistic Cancer Care

Oncology nurses play a pivotal role in delivering targeted interventions that span educational, pharmacological, psychosocial, and multidisciplinary strategies. Evidence supports nurse-led sexual health education programs, the use of validated communication models, psychosexual counseling, pharmacologic therapies, digital programs, and collaborative care as essential components of a holistic treatment approach [7,30,42,62,73,74,75]. This intervention, summarized in Table 4, has demonstrated effectiveness in improving sexual health outcomes, increasing patients’ satisfaction, and enhancing the confidence and competence of nurses in addressing this sensitive topic.

### 3.5. How to Deal with Attitudes and Beliefs of Nurses About Sexual Health

#### 3.5.1. Personal Beliefs and Knowledge Gaps

Attitudes of nurses toward sexuality significantly shape their willingness and ability to initiate sexual health discussions with cancer patients. Many nurses perceive sexuality as a private matter and express discomfort during clinical interaction, fearing that it may offend patients or be considered unprofessional [35,73]. This discomfort often stems from a lack of formal academic preparation and limited clinical exposure to sexual health topics [34,45]. For example, Jordanian students have reported feeling unprepared to address sensitive issues such as the sexual needs of terminally ill patients [59]. Instead, they often rely on informal knowledge, which may not be sufficient to manage the complexities of cancer-related sexual dysfunction. This emphasizes the need to integrate structured sexual health content into nursing curricula and continuous professional development programs [79].

#### 3.5.2. Culture and Religious Influences

In many conservative societies, sexuality remains a taboo topic, and healthcare providers, especially nurses, may avoid discussing it to conform to social expectations [37,50,69]. These cultural constraints can blur the boundaries between professional responsibilities and personal values, making it difficult for nurses to provide unbiased patient-centered care [65,70]. Additionally, patients themselves may feel too embarrassed to initiate such a discussion, instead expecting the healthcare provider to lead this conversation [35,66]. This dynamic often leads to mutual avoidance of sexual health discussion. Cultural and religious influences have a profound impact on both nurses and patients. Overcoming this requires fostering an environment that respects cultural sensitivities while promoting open, professional communication about sexuality.

#### 3.5.3. Experience, Gender, and Communication Challenges

Nurses with more experience in oncology or palliative care are more likely to feel comfortable discussing sexual dysfunction with patients [62,67]. In contrast, novice nurses often avoid this discussion due to a lack of confidence; moreover, gender dynamics further complicate communication. Nurses may feel uneasy when addressing sexual issues with patients of the opposite sex, particularly in a traditional cultural context where such interactions are considered inappropriate [39]. This discomfort intensifies when institutional policies do not support sexual health as a routine component of care. Experience plays a protective role in fostering nurse confidence, but institutional support and cultural adaptation are equally necessary to overcome gender-based communication barriers.

#### 3.5.4. Institutional Roles and Professional Development

When institutions provide clear polices, a supportive environment, and professional development opportunities focused on sexual health, nurses become more proactive in addressing these concerns [24,30,31]. Interventions such as role-playing exercises, structured communication training, and continuing education workshops have been shown to increase nurses’ comfort and perceived competence in this area [73,80]. Institutional initiatives and professional development programs are effective strategies for transforming a hesitant attitude into proactive sexual health engagement. Structured support not only improves clinical confidence but also ensures that patients receive holistic cancer care that includes their sexual well-being.

The literature reveals that while oncology nurses understand the importance of sexual health, many feel unequipped in managing related discussions because of personal discomfort, cultural influences, a lack of education, and institutional neglect. Bridging this gap requires multilevel approaches, integrating sexual health in nursing education, promoting cultural competence, offering experiential training, and embedding sexual health into routine oncology care guidelines.

#### 3.5.5. Global Effort in Oncology Sexual Health Education in Recent Years

There has been increasing global recognition of the need to improve sexual healthcare within oncology services. Academic institutions and healthcare systems in North America, in particular, have developed structured educational programs targeting oncology providers.

In Canada, the trueNTH (true North sexual health and rehabilitation e-training) program has been implemented to increase the competencies of prostate cancer care providers. A multicenter evaluation demonstrated significant improvement in participants’ sexual health knowledge and communication self-efficacy, with almost all participants reporting high satisfaction with the program [81]. Similarly, in Alberta, a dedicated oncology sexual health clinic received over 130 patient referrals in two years, identifying diagnoses of sexual dysfunction in 100% of female patients and 80% of male patients [82]. Ms. Anne Katz, based in Manitoba, should be applauded for her longstanding contributions to sexual health [83]. The Princess Margaret Cancer Center in Toronto has developed hybrid in-person and virtual sexual health services, expanding access to specialized care for a broader patient population [83].

In the United States, the Fox Chase Cancer Center launched the **iSHARE** (**i**mproving **S**exual **H**ealth and **A**ugmenting **R**elationships through **E**ducation) intervention, which focused on enhancing the ability of breast cancer clinicians to discuss sexual health interventions, leading to increased clinician confidence and sustained patient satisfaction [43].

Internationally, the 2024 summit organized by the scientific network on female sexual health and cancer exemplified a growing effort to promote interdisciplinary education and collaborative models of care [68]. These global initiatives provide compelling evidence that structured education and clinical services significantly enhance sexual health outcomes in oncology.

## 4. Discussion

In summary, this narrative review highlights that sexual dysfunction is common among cancer patients and profoundly impacts their physical health and psychological and relational well-being. Sexual side effects of cancer and its treatment (e.g., erectile dysfunction, vaginal dryness, and dyspareunia) often lead to a loss of libido, body-image concerns, and strains on intimacy [7,8,58]. Patients report high levels of anxiety and depression associated with unmet sexual needs [11,84]. Significantly, sexual health interventions (e.g., counseling, education) can ameliorate these effects. Studies have shown that targeted sexual health information reduces patients’ anxiety and improves their satisfaction and quality of life [84,85]. The review also underscores the pivotal role of oncology nurses. Nurse-led interventions (using models such as PLISSIT and BETTER) enable sensitive discussion and management of sexual concerns [46,48]. For example, structured programs for nurses have been shown to improve knowledge, confidence, and communication skills [73,86]. Canadian initiatives such as trueNTH e-training have yielded significant gains in provider knowledge and self-efficacy (with 98% satisfaction), effectively preparing nurses to support the sexual health of prostate cancer patients [71]. These interventions align with clinical guidelines: both the ASCO and the NCCN now recommend routine screening for sexual problems and timely referral for specialist care [5,34]. However, multiple barriers prevent the delivery of effective care. Across settings, nurses report a lack of knowledge and training as a primary obstacle [30,38]. Many nurses feel embarrassed or ill prepared; for example, 85% of Belgian oncology nurses and 70% of U.S. nurses in separate studies admitted to discomfort when discussing sexual issues [62,86]. Time constraints and systemic factors (no private space or protocols) are common hurdles [1,87]. Cultural and social taboos further compound these barriers. In conservative societies (e.g., the Middle East, China, and parts of Africa), discussing sexuality is considered shameful, thus discouraging both patients and providers from approaching the subject [49,77]. In Iran or Africa, women with breast cancer, for example, often feel profoundly shameful due to religious and familial norms, leading them to hide sexual problems [27,54]. Even among Western–trained nurses, cultural assumptions (e.g., patients must be grateful to be alive and not need sex) can inhibit discussion [12,26].

### 4.1. Comparisons with the Literature

These findings corroborate the findings of international nursing studies. Global surveys find that only a minority of oncology clinicians regularly address sexual health; for example, in a Latin American study, only ~10% of physicians felt adequately prepared to manage sexual issues, and ~30% rarely broached the topic [79]. The education gap noted by this review aligns with conclusions from systematic analyses: nurses worldwide recognize sexual health as important, but they report inadequate curricula and on-the-job training [1,24,25]. Recent evidence from the past few years further reinforces these points. Concept mapping in Hong Kong highlighted the need for culturally adapted care models (an extended PLISSIT) to fit the local practice environment [52]. The current adult survivorship guidelines of the National Comprehensive Cancer Network of the United States (NCCN) in 2024 explicitly advocate routine sexual function screening and multidisciplinary referral, reflecting an international shift toward the integration of sexual rehabilitation in cancer [5,34]. Recent oncology nursing studies have reinforced the notion that targeted interventions can make a significant difference. For example, a scoping review revealed that a brief active–learning workshop (e.g., role-play using PLISSIT scenarios) significantly improved nurses’ communication comfort and knowledge [1,73,86]. Novel e-learning modules (such as those for adolescent/young adult cancer patients) have been shown to increase awareness of reproductive and sexual health issues among oncology staff [53,57]. Notably, the current review’s Middle East context aligns with new data: a 2025 Jordanian survey reported similar barriers (time, training, and culture) and called for system-level educational reforms [1]. Conversely, research from countries with dedicated sexual health clinics (e.g., Canada’s multidisciplinary clinics) suggests that nurse involvement in such teams leads to more diagnoses and interventions for dysfunction, again supporting the central role of nursing [71,88]. Overall, these additional studies corroborate our narrative review findings and highlight growing global momentum toward addressing cancer-related sexual dysfunction as a standard component of care. This review adds to the current literature in a timely manner by summarizing current information and discussing practical solutions to the growing global problem, as more patients survive initial cancer treatment to address posttreatment sexual health complications.

### 4.2. Implications for Practice, Policy, and Education

Given these insights, several practical implications emerge. Clinically, oncology nurses should be empowered to initiate sexual health conversations routinely. Models such as the 5 A’s (Ask, Assess, Advise, Assist, Arrange) and structured frameworks (PLISSIT/BETTER) provide practical guides for nurses to normalize discussions and offer basic interventions [46,47,89].

Nurses and nurse practitioners can help mitigate physical side effects through anticipatory guidance—for example, by recommending lubricants, topical vaginal estrogen creams, vaginal dilators, and pelvic floor exercises. They can also connect patients to appropriate specialists for more complex issues, such as sex therapists, fertility counselors, and urologists [80,90]. For instance, nurses can collaborate with other team members, including family physicians and pharmacists, to select the most suitable phosphodiesterase inhibitor for an individual patient, taking into account cardiac comorbidities and lifestyle factors. In general, sildenafil acts more quickly but has a shorter duration of effect compared to tadalafil.

Additionally, nurses can refer patients to plastic surgeons for breast or testicular prostheses. Some men may experience distress following orchiectomy, and the insertion of a testicular prosthesis comparable in size to the contralateral testicle can significantly improve body image.

The evidence suggests that even brief interventions by nurses providing permission, education, and simple suggestions can significantly improve sexual function outcomes (up to 70% of patients can return to baseline with intervention [7,66,91]). Thus, oncology units should allocate time and privacy for these assessments [86].

In terms of policy, institutions and professional organizations must prioritize sexual health. Cancer care guidelines should formally include sexual function as a survivorship quality metric. For example, the NCCN now lists “sexual health” under long-term effects in its survivorship guidelines [5,34]. Hospitals could adopt standardized sexual history checklists in electronic records and track sexual health referral rates as a quality indicator [71]. Reimbursement and scheduling policies should recognize sexual counseling as part of patients’ care. In culturally conservative contexts, policy and administration can facilitate gender-sensitive care (e.g., ensuring the availability of female providers for intimate issues) and provide private settings for discussions.

National health bodies might consider public education campaigns to destigmatize post-treatment sexual issues, as in some Western countries [49,77]. For nursing education, curricula must be revamped. Both pre-licensuring and continuing education programs should incorporate content related to sexual health, anatomy, communication skills, psychosocial aspects, and cultural competence. Simulation and role play (which nurses find highly effective) can prepare nurses to handle these sensitive topics [28,73]. Promising training programs, such as **ENRICH** (**E**ducating **N**urses about **R**eproductive **I**ssues in **C**ancer **H**ealthcare, a web-based program on fertility/sexuality training for pediatric oncology nurses), should be expanded to all oncology specialties [57,71]. Academic nursing programs in conservative societies should emphasize that discussing sexuality is integral to holistic cancer care and teaching students to use neutral, validating language. Professional development initiatives (workshops, online modules) are needed to close the current knowledge gap. One review explicitly recommended the creation of oncology nurse training programs to improve attitudes and skills related to sexual health [29,42]. Collectively, these steps can transform the culture of care so that sexual well-being is routinely addressed.

### 4.3. Strengths and Limitations of the Review

This narrative review offers a broad synthesis of nursing roles in sexual dysfunction, drawing on diverse sources across cultures and practice settings. Its strengths include integrating recent evidence (e.g., studies from 2024 to 2025) and highlighting underrecognized factors (such as cultural taboos in conservative societies) [1,77]. Focusing on nursing perspectives, this review fills a gap left by many reviews, mostly written for medical doctors [5,53]. The authors’ experiences in the Middle East and China provide valuable insights into taboo barriers. The reference section includes the most important recent studies, and this review is poised to provide valuable guidance for clinicians, nurses, and researchers.

As a narrative (rather than systematic) review, it is subject to specific limitations. Unlike a systematic review, the search strategy may not have been comprehensive; narrative syntheses do not require appraisal. Important studies might have been missed, and the selection of literature can reflect author bias [25,55]. Additionally, without a standardized methodology, reproducibility may be limited. The review mentions primarily English-language and peer-reviewed sources, potentially omitting the other relevant literature or non-English research (e.g., local studies in Asia/Africa) [29,53]. Some of the studies suffer from cultural inhomogeneity, a lack of longitudinal studies, and/or poor methodological quality. Finally, without a meta-analysis, the review provides only a qualitative conclusion regarding different settings in low- versus high-income countries and different cultures. Table 5 summarizes this perspective. Thus, the findings should be interpreted as reflective insights grounded in expert synthesis rather than conclusive empirical evidence.

### 4.4. Recommendations and Future Directions for Research

Future research should evaluate and refine specific nursing interventions to increase their effectiveness. Randomized trials or controlled studies could test the impact of nursing-led sexual counseling on patient outcomes [46,47]. Comparative research should identify which educational methods (e.g., in-person workshops vs. online modules) are most effective in improving nurses’ competence [73,86]. Culturally tailored models deserve study, for example, by assessing interventions that respect social norms in Middle Eastern or African contexts [49,77]. Research gaps include male sexuality (most studies focus on women) and underserved groups, such as low-income or LGBTQ (Lesbian, Gay, Bisexual, Transgender, Queer or Questioning) patients [7,87]. Health services research could explore how institutional changes (e.g., establishing oncology–sexuality clinics) impact the quality of care [71,88]. Education and training initiatives must be expanded. Oncology units should implement routine sexual health training for all healthcare staff, with refresher sessions to maintain skills [52,57].

## 5. Conclusions

Empowering nurses through structured education, standardized guidelines, and open communication strategies is essential for improving patient outcomes. Addressing sexual dysfunction as a routine aspect of cancer care will enhance survivors’ quality of life and foster a more holistic approach to oncology treatment.

This research was conducted by a team of researchers with origins in the Middle East and China, both of whom represent conservative traditions and diverse religious backgrounds. This concise overview, which is rich in detailed references and clinical pearls, offers a unique and highly educational resource for healthcare professionals across multiple disciplines, such as nurses, physicians, social workers, psychologists, music therapists, sex therapists, and chaplains. It therefore has broad clinical implications, and practical suggestions will greatly benefit cancer patients and their providers from different disciplines.

## Figures and Tables

**Figure 1 curroncol-32-00457-f001:**
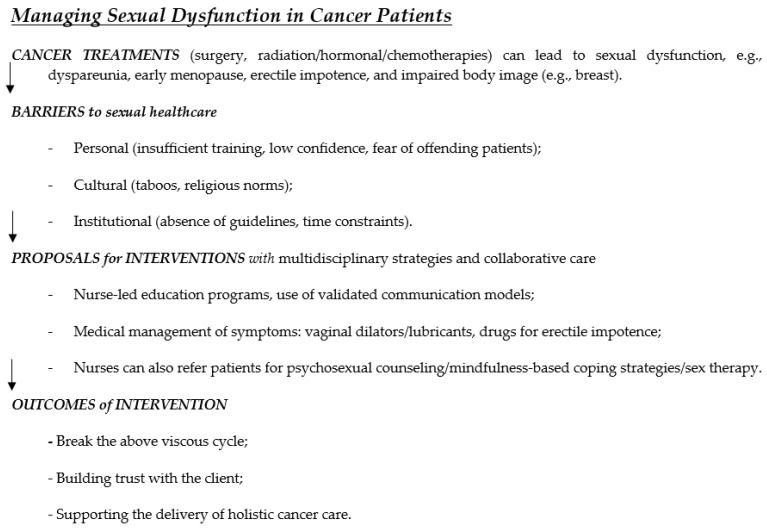
The conceptual framework summarizes barriers, interventions, and outcomes related to nursing care in individuals with sexual dysfunction.

**Table 1 curroncol-32-00457-t001:** Thematic summary.

Thematic Area	Summary Point	References
**Prevalence and impact of sexual dysfunction**	Sexual dysfunction is highly prevalent (affecting approximately 50–70% of survivors). It has a significant negative impact on quality of life and intimate relationships, yet it often remains underrecognized in oncology practice. Addressing this issue requires integrating sexual health into routine survivorship care.	[18,24,25,26,27,28,29]
**Psychological and social impact**	Untreated sexual dysfunction leads to psychological distress (e.g., anxiety, depression) and social problems such as relationship strain and isolation. A lack of open communication about sexual issues can exacerbate these psychological outcomes, indicating a need for holistic support for patients and partners.	[24,30,31,32,33]
**Barriers in cancer care**	Oncology nurses face multifaceted barriers–personal (insufficient training, low confidence), cultural (taboos, religious norms), and institutional (absence of guidelines, time constraints)– that hinder discussions of sexual health. These barriers contribute to a gap in care, as many patients’ sexual concerns go unaddressed.	[34,35,36,37,38,39,40]
**Interventions for managing sexual dysfunction**	A range of nursing interventions can improve sexual health outcomes: patients’ education and counseling (including models like PLISSIT and BETTER), appropriate use of medications (e.g., PDE5 inhibitors for erectile dysfunction, vaginal moisturizers or estrogen for dyspareunia), and multidisciplinary approaches (involving oncologists, psychologists, and sex therapists). Implementing these interventions—alongside nurse training programs and sexual health clinics—has been shown to increase patients’ satisfaction and sexual functioning. Clinical guidelines now recommend proactive sexual rehabilitation as part of standard cancer care.	[7,24,30,41,42,43]
**Nurses attitudes and beliefs**	Nurses’ attitudes and beliefs can significantly impact sexual healthcare. Many nurses feel uncomfortable or believe sexual matters are too private to discuss, often due to cultural/religious influences or lack of training. Such attitudes can prevent essential conversations, training, and education. Hence, its integration into nursing curricula and professional development, along with supportive institutional polices, is crucial to change perceptions and empower nurses to address sexual dysfunction confidently.	[30,34,35,36,37,44,45]

BETTER: B: bring up the topic, E: explain that you are concerned with all aspects of patients’ lives, T: tell patients that sexual dysfunction is common, T: timing might not be right now, but you are open to future discussion, E: educate patients about sexual side effects, R: record the conversation in the medical record; PDE5: phosphodiesterase type (inhibitors); PLISSIT: P: permission, LI: limited information, SS: specific suggestions, IT: intensive therapy.

**Table 3 curroncol-32-00457-t003:** Summary description of barriers to addressing sexual dysfunction.

Barrier	Description	References
Provider discomfort and training deficits	Many healthcare providers lack specific sexual health training and feel anxious or unprepared to discuss intimacy topics. This leads to the avoidance of sexual health conversations to prevent offending patients.	[34,43,68]
Cultural and religious norms	In many cultures, sexual topics are taboo. Providers fear violating social norms and may face adverse reactions. Such taboos create strong reluctance for both providers and patients to bring up sexual issues.	[36,37,45,69,70,71]
Institutional constraints	Hospital often has no standardized protocol or guidelines for addressing sexual health, and nurses have limited time in busy oncology settings. Lack of institutional support (e.g., training resources, specialized experts) further hinders the integration of sexual health into routine care.	[24,30,38,39,40,66,72]
Patient-related factors	Patients themselves frequently feel embarrassed or uncertain about the relevance of sexual health issues and may prefer providers to initiate the topic. Many reports indicate that it is easier to discuss sexual concerns with a same gender provider, and both sides often wait for the other to speak first, resulting in unmet needs.	[24,40,62,65,72]

**Table 4 curroncol-32-00457-t004:** Summary descriptions of interventions for managing sexual dysfunction in cancer patients.

Intervention	Description	References
**Nurse-led education and counseling**	Structured sexual health education and counseling sessions (e.g., BETTER Model) help patients understand treatment-related changes and coping strategies. A nurse-led program improves patients’ knowledge about the physical and emotional aspects of sexuality after cancer.	[7,44,74]
**Pharmacological treatment**	Medication can directly address physical dysfunction, e.g., phosphodiesterase-5 inhibitors (sildenafil, tadalafil) improve post-treatment erectile function, and topical estrogen therapies with vaginal moisturizers relieve vaginal atrophy and dyspareunia. Nurses play a key role in informing patients about this treatment and managing side effects.	[41,42,53,66,75]
**Psycho-Sexual therapy**	Individual and couple counseling (psychosexual therapy) addresses the emotional and relational impact of sexual dysfunction. Clinical studies show that such counseling improves sexual satisfaction, reduces anxiety, and enhances partners’ communication. Nurse-led psychosexual support provides emotional assistance along with practical suggestions for intimacy.	[30,31,42,62]
**Multi-disciplinary team care**	Collaborative care involving oncologists, nurses, psychologists, physiotherapists, and sex therapists creates comprehensive treatment plans. Establishing dedicated sexual health clinics within oncology departments has been shown to improve patients’ outcomes and satisfaction.	[24,30,62,73]
**Digital and Telehealth intervention**	Telemedicine platforms, online education modules, and virtual support groups offer accessible sexual health resources. These can overcome geographic and stigma barriers, allowing confidential counseling and education. Studies find digital programs help patients engage with sexual healthcare despite time or cultural constraints.	[44,75,76,77]
**Nurse training and frameworks**	Targeted training programs and communication frameworks (e.g., PLISSIT or stepped-skill models) build nurses’ confidence and skills. Education and systems-based strategies (workshops, protocols) are shown to increase their competence in discussing sexual concerns and conducting more proactive and routine sexual health assessments.	[7,24,30,45,62,74,75,78,79,80]

BETTER: B: Bringing up the topic. E: Explain that you are concerned with all aspects of patients’ lives. T: Tells patients that sexual dysfunction is common. T: Timing might not be right now, but you are open to future discussion. E: Educate patients about sexual side effects. R: Record the conversation in the medical records; PLISSIT, P: Permission. LI: Limited Information. SS: Specific Suggestions. IT: Intensive Therapy.

**Table 5 curroncol-32-00457-t005:** Comparative overview of how nursing roles in addressing sexual dysfunction among oncology patients differ between high-income countries and low- and middle-income countries.

	High-Income Countries	Low- and Middle-Income Countries
Training and education	Specialized training programs like TrueNTH (Canada) and iSHARE (USA) enhance nurses’ confidence and skills in sexual health communication.	Limited access to formal training; nurses often lack education in sexual health and oncology-specific sexual dysfunction.
Clinical guidelines	National guidelines (e.g., ASCO, NCCN) recommend routine screening and referrals for sexual dysfunction.	Guidelines are often absent or not enforced; sexual health is rarely integrated into oncology care plans.
Multidisciplinary clinics	Dedicated sexual health clinics exist within cancer centers (e.g., Alberta, Manitoba in Canada).	Few or no specialized clinics; care is fragmented and often deprioritized due to resource constraints.
Cultural acceptance	Increasing openness to discussing sexual health; use of structured models like PLISSIT and BETTER.	Cultural taboos and stigma inhibit open discussions; patients and providers may avoid the topic entirely.
Patient engagement	Patients are encouraged to voice concerns; nurses initiate conversations during survivorship planning.	Patients rarely disclose sexual concerns due to embarrassment, stigma, or belief that dysfunction is inevitable.
Access to resources	Availability of lubricants, dilators, sex therapy referrals, and pelvic floor physiotherapy.	Limited or no access to sexual health resources; interventions are often nonpharmacological due to cost.
Technology and innovation	Using online modules, telehealth, and digital tools for training and patient support.	Minimal technology use; infrastructure and internet access may be barriers.

ASCO, American Society of Clinical Oncology; BETTER, B: Bringing up the topic. E: Explain that you are concerned with all aspects of patients’ lives. T: Tells patients that sexual dysfunction is common. T: Timing might not be right now, but you are open to future discussion. E: Educate patients about sexual side effects. R: Recorded the conversation in the medical records; iSHARE, improved Sexual Health and Augmented Relationships through Education; NCCN, National Comprehensive Cancer Network; PLISSIT, P: Permission. LI: Limited Information. SS: Specific Suggestions. IT: Intensive Therapy; TrueNTH: True North Sexual Health and Rehabilitation eClinic.

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
