# Peer review of "A Narrative Review of the Roles of Nursing in Addressing Sexual Dysfunction in Oncology Patients"

_curroncol, 2025, doi:10.3390/curroncol32080457_

Round 1

Reviewer 1 Report

Comments and Suggestions for Authors

*Overall Peer Review Comment:

This narrative review addresses a crucial and often overlooked issue—sexual dysfunction in oncology patients—with an emphasis on the role of nurses. The manuscript is timely, comprehensive, and well-supported by a diverse range of references. It effectively synthesizes data on prevalence, barriers, and interventions. However, it would benefit from improved clarity, better structural coherence, and a more critical discussion. Enhancing the language style, reducing redundancy, and organizing the content more effectively would further strengthen the manuscript’s academic rigor and practical utility.

*Section-by-Section Comments

  1. Title

Evaluation:

-The title effectively reflects the content and scope of the paper.

Suggestions:

-Consider rephrasing for improved fluency:

-Original: “A Narrative Review on The Roles of Nursing in Sexual Dysfunction Among Oncological Patients”

-Revised: “A Narrative Review of Nursing Roles in Addressing Sexual Dysfunction in Oncology Patients”

-This revision enhances readability and aligns better with academic conventions.

  1. Abstract

Evaluation: 

-The abstract effectively summarizes the paper's purpose, methods, findings, and implications. 

Suggestions: 

- Avoid redundancy; for example, the phrase “underdiagnosed and undertreated” is repeated later in the text. 

- Include specific results or themes identified, such as training, stigma, and multidisciplinary collaboration. 

- Conclude with a sentence addressing future directions or research implications. 

  1. Introduction

Evaluation: 

- The introduction provides a clear rationale and is supported by strong citations. 

- The prevalence and significance of the topic are well-supported. 

Suggestions: 

- Some sections contain excessive repetition. 

- Consider organizing the content into sub-paragraphs with a more transparent structure: prevalence → impact → role of nurses → existing gaps → aim. 

- Ensure proper citation formatting; for example, the WHO should not be referenced with the year in brackets and again at the end of the sentence. 

  1. Methods

Evaluation: 

-The methods section outlines the databases used and the criteria for inclusion and exclusion; however, it lacks sufficient depth and transparency. 

Suggestions: 

- Provide a more detailed description of the search terms used. 

- State the total number of articles that were screened and the number that were ultimately selected. 

- Justify the decision to conduct a narrative review instead of a systematic review.

  1. Results

Evaluation:

- The five summarized categories are relevant.

- The content is rich and well-supported by literature.

Suggestions:

- Reorganize the sections into clearly numbered or bullet-pointed subheadings to enhance readability.

- Include a descriptive summary table for each intervention or barrier.

- Add synthesis or interpretation at the end of each subsection to avoid giving the impression of simply listing literature.

  1. Discussion

Evaluation: 

-Many vital issues are appropriately discussed, including stigma, cultural factors, training, and the psychological burden involved.

Suggestions: 

-The discussion could be strengthened by providing a critical analysis and synthesis of findings across various studies. It's important to highlight differences in settings, such as between high-income and low-income countries, as well as the varying cultural contexts. Additionally, clearly stating the implications for nursing practice, policy, and future research would enhance the overall impact of the discussion.

  1. Conclusion

Evaluation: 

- The conclusion emphasizes the key messages of the article. 

Suggestions: 

- Avoid restating the abstract; instead, synthesize the findings and provide concrete recommendations. 

- Consider acknowledging the limitations of the review (e.g., narrative format, language bias).

  1. Figures

Evaluation:

- No figures are included.

Suggestions:

- Include a conceptual framework figure that summarizes barriers, interventions, and outcomes related to nursing care in sexual dysfunction.

- Adding visuals, such as a flowchart of proposed interventions, would enhance clarity and engagement.

  1. Tables

Evaluation: 

- Table 1 is informative but too condensed. 

Suggestions: 

- Expand the table to include examples, prevalence rates, and interventions for each domain. 

- Use more transparent labels and add references to each row for transparency.

  1. References

Evaluation:

- A wide range of current and relevant references has been cited.

Suggestions:

- Ensure consistency in citation formatting, as some references are missing proper journal names or DOIs.

- Remove any duplicates (e.g., WHO 2022 being cited multiple times with the duplicate content).

- Adopt a consistent referencing style, likely APA or Vancouver, depending on the journal's guidelines.

*Recommendation 

Minor to moderate revisions are needed before acceptance. This paper addresses a significant gap in cancer care and offers valuable recommendations. With better organization, a more critical analysis of the literature, and polished writing, the article has the potential to make a meaningful contribution to the field.

Reviewer 2 Report

Comments and Suggestions for Authors

Abstract

The abstract only mentions a “comprehensive literature review” without details: missing databases, period, type of studies included, selection criteria. It is not indicated whether narrative review guidelines were used

It is stated that sexual dysfunction is prevalent, but no specific numbers or references to the population analysed are given. No reference to the number of studies included, no summary of key findings. What is already known is repeated (‘integrating sexual health care is essential’), without offering a specific result derived from the review. Expressions such as ‘pivotal role’ or ‘empower nurses’ are vague. A clear statement of novelty or practical implications is lacking.

Introduction

The first paragraph opens with a too general and vague sentence (‘Sexual dysfunction remains a common untreated medical issue among cancer patients...’). The second sentence states that “this work is part of our published project”, which is out of place in a scientific introduction. Concepts such as “impacts on quality of life, intimate relationships, mental health” are repeated up to 3 times with different words. The WHO is mentioned several times with similar information (2014, 2009, 2022) without critical development. It is not explicitly stated which gap in the literature is to be filled. A sentence such as: ‘This review aims to answer...’ or ‘To our knowledge, no narrative reviews have...’. Some statements are too generalised: e.g. “sexual dysfunction is independent of gender” should be qualified. Others are inaccurate: ‘hurting desire while blocking arousal’ is not appropriate clinical wording. For being a review on the role of the nurse, the introduction devotes very little space to nursing: it is mentioned in a generic and secondary way.

Methods:

The section is vaguely and imprecisely described. Inclusion and exclusion criteria are not specified, nor how the results were selected and summarised. No formalised narrative review strategy (e.g. SANRA, PRISMA-Scr, etc.) is explicit. The databases listed are numerous, but the search strategy is not made explicit, nor are keywords, Boolean strings or the number of results found.

Results

The section does not present an ordered summary of the results of the review. There is no table with included studies or characteristics of the articles analysed. Table 1 is only a thematic list disguised as a table, presenting no authors, years, methods, samples or summary results. The section is descriptive and discursive, with no comparative analysis between studies. There is no stratification by type of sexual dysfunction, cancer site, gender, or health context.

Prevalence information is repeated with minimal variation. Sub-themes (e.g. psychological impact, cultural barriers, educational approaches) are not organised as emergent categories, but appear mixed. The numbers reported (e.g. ‘60-90% of prostate cancer patients develop erectile dysfunction’) are not clearly linked to the sources. It is not always clear which study supports which claim, and citations are sometimes excessive and not selective.

Discussion

The section seems to repeat the contents of the results, rather than interpret them. There is a lack of critical reflection on the limitations of the included studies (e.g. cultural inhomogeneity, lack of longitudinal studies, poor methodological quality). Gaps in the literature or areas for further investigation are not identified. Lack of comparison with other works or systematic reviews There is no discussion of whether other reviews exist on the same topic, nor how this work differs. The article concludes that ‘educational programmes improve nursing effectiveness’, but provides no systematic evidence or evaluation of effectiveness. Some statements seem opinion-based rather than evidence-based. There is a lack of proposals applicable in clinical practice: e.g. nursing checklists, short protocols, educational pathways. Absence of limitations of the review

Comments on the Quality of English Language

The manuscript would benefit from a careful English language revision. Some sentences are unclear or repetitive, and certain expressions are not appropriate for scientific writing. A professional editing service or review by a fluent English speaker is recommended to improve clarity and flow.

Round 2

Reviewer 2 Report

Comments and Suggestions for Authors

I would like to thank the authors for their thorough revision and detailed response. The manuscript has been greatly improved in terms of structure, methodology, and language. The abstract now includes relevant information; the introduction is more focused and coherent; the methods are clearly described and justified; the results are better organised and accompanied by informative tables. The discussion is more analytical, with practical implications better explained and greater attention to socio-cultural contexts. The conclusion is now more incisive and comprehensive.
I suggest only a few minor changes: standardise the table headings, make some examples in the clinical implications more concrete, and further refine the language in some sentences.

Comments on the Quality of English Language

The overall quality of the English language has improved considerably compared to the previous version. The manuscript is now clear and generally well-written. Minor grammatical and stylistic inconsistencies remain in a few sections, and some expressions could benefit from further refinement. A final proofreading by a native English speaker or professional editor is recommended to ensure fluency and precision throughout the text.
